# Pediatric Diabetic Ketoacidosis (PDKA) among newly diagnosed diabetic patients at Dilla University Hospital, Dilla, Ethiopia: Prevalence and predictors

Dinberu Oyamo Oromo [ID]*

Department of Pediatrics and Child Health, College of Health Sciences, Dilla University, Dilla, Ethiopia

* dinbe80@gmail.com

## Abstract

### Background

Diabetic ketoacidosis (DKA) is a morbid complication of Type 1 diabetes mellitus(T1DM), and its occurrence at diagnosis has rarely been studied in Ethiopia, despite the many cases seen in the pediatric population.

### Objective

The aim of this study was to know the prevalence of DKA among patients with newly diagnosed diabetes mellitus and identify avoidable risk factors.

### Method

This institution-based retrospective cross-sectional study was conducted from December 1, 2018 to December1, 2022. Newly diagnosed T1DM under 15 years were included in the study. DKA and the new diagnosis of type 1 DM were defined based on the 2022 ISPAD and other international guidelines. A data collection form was used to collect sociodemographic and clinical data. Descriptive, bivariate, and multivariate logistic regression analyses were conducted to identify the risk factors.

### Result

Among the 61 newly diagnosed T1DM pediatric patients admitted, DKA was the initial presentation in 37 patients, accounting for 60.7% of the cases. The mean age at diagnosis was 8 (±3.85) years, with females being more affected. Clinical presentation revealed vomiting accompanied by signs of dehydration (32.4%), with polyuria, polydipsia and weight loss (26.2%) being the most common symptoms. The presence of adequate knowledge of signs and symptoms of DM (AOR = 0.07, 95%CI 0.019–0.0897, P value 0.017) and a family history of DM (AOR = 0.129 95%CI 0.019–0.897, P value 0.039) were protective factors against DKA as the initial diagnosis of DM. Moreover, new-onset type 1 DM without DKA was 1.5 times higher in children from families with a high monthly income (AOR = 1.473,

**Funding:** The author(s) received no specific funding for this work.

**Competing interests:** The author has declared that no competing interests exist.

95% CI 0.679–3.195 p value 0.000) compared to those from families with low income. The presence of an infection prior to DKA (AOR = 11.69,95%CI 1.34–10.1,P value 0.026) was associated with the diagnosis of DKA at the initial presentation of DM.

## Conclusion

A high number of children present with diabetic ketoacidosis (DKA) at the initial diagnosis of diabetes mellitus (DM), which is associated with inadequate knowledge of the signs and symptoms of DM as well as the masking effect of concomitant infections in these children. Healthcare professionals should endeavor to suspect and screen children. Continuous awareness creation of DM is encouraged to diagnose diabetes mellitus earlier and to decrease the prevalence of DKA as an initial presentation.

## Introduction

Diabetic ketoacidosis (DKA) is a common complication of type 1 diabetes mellitus (T1DM) in both children and adolescents [1, 2]. Worldwide, the reported frequencies of DKA as an initial presentation of type 1DM across countries and continents are diverse ranging from as low as 12% and as high as 80%, indicating wide geographic variations in its incidence [3].

This condition remains a public health concern in sub-Saharan African countries, including Ethiopia [4]. A Small number of reports from Africa have reported that up to 95% of children present with DKA at the time of type 1DM diagnosis [3, 5, 6]; and in Ethiopia, there were reports showing its high prevalence [7–9].

Earlier diagnosis of diabetes mellitus in children and adolescents will reduce the number of children presenting with DKA. However, several studies have demonstrated and linked an increased incidence to multiple factors. For instance, younger age [10–15], uneducated parents, no relative with type1DM, residing in rural areas, and having low economic status were incriminated [10, 13, 14, 16]. Nonetheless, not all the factors were explanatory [3].

Considering the fewer reports in Ethiopia, the current study was conducted to know the prevalence of DKA among newly diagnosed type 1DM children and to uncover potential factors for its occurrence as a first presentation.

## Participants and methods

### Study area

The study was conducted at Dilla University Hospital, located in Dilla town, southern Ethiopia, 360 km away from the Ethiopian capital city, Addis Ababa. The hospital is organized into different departments, and the pediatric and child health departments have emergency, inpatient, outpatient, and neonatal units. The pediatric ward admits an average of 875 pediatric patients for inpatient treatment services every month.

### Study design and patient selection

The study utilized an institution-based retrospective cross-sectional method over a 4-year period, from December 1,2018 to December 1,2022. Data were accessed and collected from individual patient charts, and as well as patients' parents/guardians from December 5,2022 to December 25,2022. All newly diagnosed T1DM patients under the age of 15 were included,

and the percentage of patients with DKA was calculated. Known T1DM patients who were admitted during the study period were excluded.

## Data collection method and study variables

Newly diagnosed T1DM patients were identified using an inpatient registration book, and their individual patient charts were retrieved from the card room for data collection. A paper based structured data collection form was prepared to gather data from individual patient charts, and parents were also contacted and interviewed for additional details. The data were collected by a health care professional working in the hospital after receiving training. The collected data were double-checked by the investigator daily for completeness and accuracy. The structured data collection form included variables such as socio-demographics (age, sex, family income, parental employment status, parental educational status, and parental marital status), parental knowledge of DM signs and symptoms, family history of DM-specific symptoms (polyuria, polydipsia, polyphagia and weight loss), preceding signs and symptoms of DM/DKA,and the presence of any preceding infections. The information was pretested on 5% of the patient records, and appropriate changes were made to refine the quality of the tool.

## Definitions

Diabetic ketoacidosis (DKA) diagnostic Criteria includes the following: hyperglycemia (blood glucose $>$ 200mg/dl), acidosis (venous pH $<$7.3 or serum bicarbonate $<$18 mmol/L) and the presence of ketonuria ($\geq$+2) or ketonemia. The degree of acidosis is used for severity assessment [1]. In resource-limited settings, such as ours, a modified diagnostic and treatment protocol was adopted from the International Society for Pediatric and Adolescent Diabetes (ISPAD) and other international guidelines [17–20]. Under this protocol, DKA was diagnosed using the following criteria: hyperglycemia of $>$200mg/dl, glycosuria, and ketonuria ($\geq$+2). Clinical severity assessment and treatment are based on the following features: mental status changes (ranging from alertness to coma), levels of dehydration (no dehydration, some dehydration and severe dehydration) and Kussmaul respiration. If DKA is not present, Type 1 DM is diagnosed when a child exhibits classic DM symptoms such as polyuria, polydipsia, polyphagia, and weight loss, along with either plasma random blood glucose level of $\geq$200mg/dl or fasting plasma glucose level of $\geq$126mg/dl.

Parental awareness of Diabetes Mellitus was evaluated through interviews focusing on knowledge of DM specific symptoms. Parents who could identify the three classic symptoms (polyuria, polydipsia, and polyphagia), along with weight loss were deemed knowledgeable, while those who could not were classified as not knowledgeable.

A preceding infection was considered if it occurred in the two weeks prior to the diagnosis of DKA. Signs and symptoms indicative of DM/DKA prior to DKA include a child exhibiting polyuria, polydipsia, and weight loss within the same two-week time frame.

A Family history of a first-degree relative with DM refers to the child's father, mother or any other siblings diagnosed with DM.

Family income in Ethiopia is classified according to the World Bank's designation as low-income nation. Local household income is categorized based on economic conditions, cost of living, and regional differences (urban and rural) over time. For the study period (2018 to 2022), family income was classified in Ethiopian birr (USD) as follows:

1. Low-income class: monthly income of less than 1000 ($<$18.02)

2. Lower-middle-income class: monthly income of 1001–2200 (18.02–39.64)

3. Upper-middle-income class: monthly income of 2201–3400 (39.65–61.62)

4. High-income class: monthly income of greater than 3401–3600 (61.63–64.87)

Parental employment status reflects the source of income and was assessed based on the following classifications: unemployed, employed civil servants, daily labor workers, and multiple sources of income apart from those mentioned.

Parental education status was assessed based on the Ethiopian Ministry of Education's educational classification as unable to read and write, can only read and write, primary education (grades 1–8),secondary education(grades 9–12) and higher education(above grade 12,including attendance at colleges and universities).

## Data analysis

Data were checked for completeness, coded, and entered into Epi Info v7 and SPSS version 21. Outliers were identified, checked accordingly, and transformed into categorical variables for correction. Data are reported as mean, median, and standard deviation for continuous variables, and counts and percentages for categorical variables.

The associations between independent variables (age, sex, family income, family history of DM, infection/acute febrile illness before presentation, Preceding signs and symptoms of DM/ DKA, parental employment, parental educational status, and knowledge of signs and symptoms of DM) and the dependent variable (DKA in newly diagnosed type 1 DM) were assessed using binary logistic regression analysis. Variables that showed an association with the outcome variable (p value of $<0.25$) were selected for multivariate analysis to control for potential confounders.

Model fitness was tested using the Hosmer-Lemeshow goodness of fit, and adjusted odds ratio (AOR) and 95%confidence interval (CI) were estimated to assess the strength of association. Statistical significance was set at a p-value of $<0.05$.

## Ethical clearance and consent to participate

Before conducting the research, ethical clearance was obtained from the Dilla University College of Health Sciences Institutional Review Board on 3/12/2022, ID number 005/17. Verbal consent was obtained from parents/guardians after explaining the purpose and details of the research and informing them of their full right to participate or withdraw at any time. Permission was also obtained from Dilla University Hospital to collect data from patient charts; consent was not needed for this. Data were collected while ensuring patient confidentiality and removing any identifiers.

## Results

During the study period (four years), 61 newly diagnosed type 1DM pediatric patients were admitted for inpatient treatment. Diabetic ketoacidosis (DKA) was the initial presentation in 37 patients, accounting for 60.7% of newly diagnosed type 1 DM cases. Of the 61 children, more than half (60.7%) of the patients were females, with a male-to-female ratio of 1:1.54 (Fig 1).

It was observed that females (n = 24, 64.8%) presented with DKA more frequently than did men (n = 13, 35.2%). Both sexes had similar age distributions, with a mean age of 8(±3.85) years. Regarding the marital status of the child parents/guardians of the 61 children, 53(86.9%) were married, five (8.2%) were single, one (1.6%) was widowed, and the remaining were divorced. The majority of parents (n = 24, 39.3%) earned above 64.87 USD, where the majority of the parents/guardians were either employed as civil servants or daily laborers (4(6.6%) unemployed, 22(36.1%) civil servants, 17(27.9%) daily laborers and other 12 (19.7) have

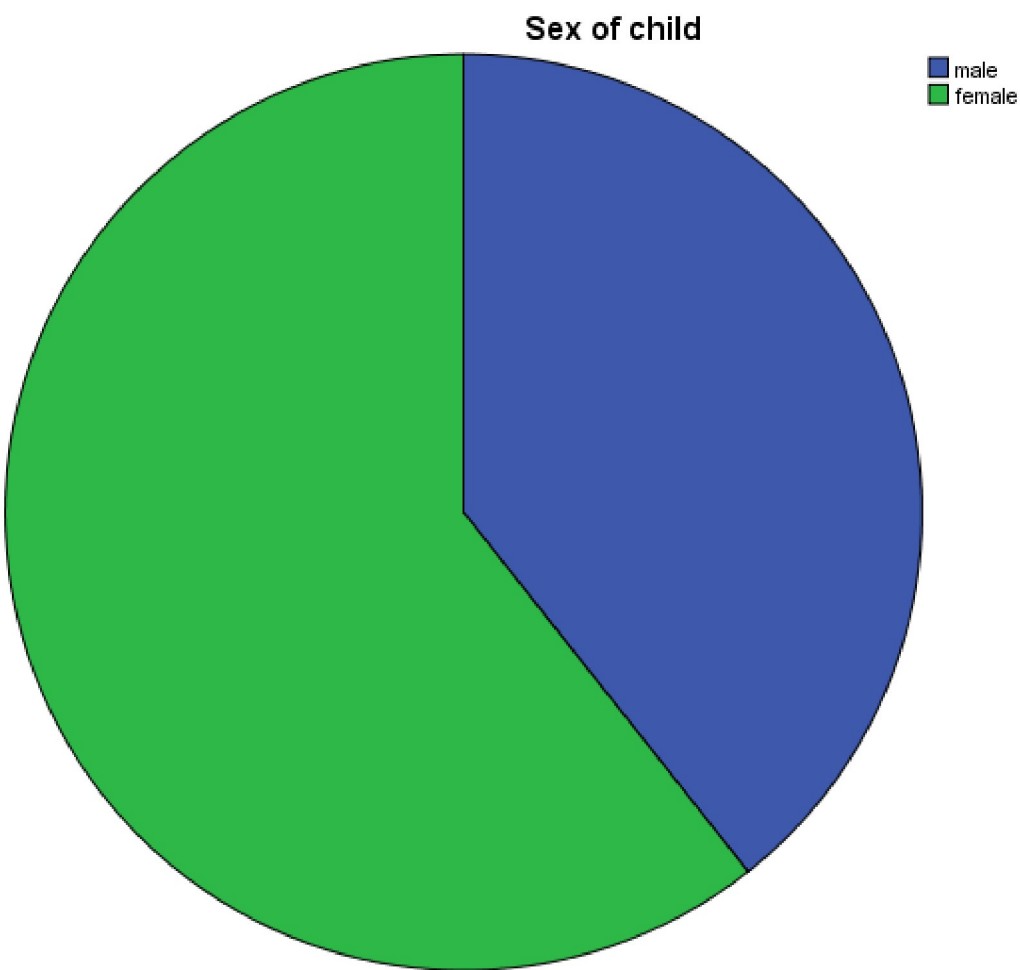

**Fig 1. Sex distribution of newly diagnosed type 1 Diabetes Mellitus (DM) from 2018 to 2022.**

multiple sources of income). Educational levels of the mothers showed that 19(31.1%) can only read and write, 19(31.1%) were in primary education, 16(26.2%) were in secondary education and only 5 (8.2%) of them were in higher education. Similarly paternal education level showed 11(18.2%) can only read and write, 14(23%) were in primary education, 16(26.2%) were in secondary education, and 20(32.8%) were in higher education (Table 1).

The majority, 41 (67.2%), of the children did not have a family history of diabetes, and 33 (54%) parents knew the signs and symptoms of DM. The clinical presentations of the 37 children with DKA were abdominal pain in 10(27.2%), loss of consciousness in 4(10.8%), vomiting accompanied by dehydration in 14(37.8%) and fast breathing in the remaining 9(29.7%). Polyuria, polydipsia accompanied by weight loss were the most common signs and symptoms of type 1 DM prior to the diagnosis of DM. Infection occurred in 26 children, 21 of whom had DKA. Of the 21 infections in DKA, pneumonia was the most commonly diagnosed infection in seven (33.3%) children and diarrheal disease in six (33.3%), followed by urinary tract infection in four (19%), and acute tonsillar pharyngitis in three (14.3%) (Tables 1 and 2).

In the bivariate analysis, family history of DM (p = 0.006, OR = 0.197 CI(0.063–0.623), family income (p = 0.00, OR = 2.26(1.437–3.55), parents' knowledge of signs and symptoms, p value 0.00, OR = 0.069 CI(0.017–0.276), and preceding infection, OR = 4.987 CI(1.532–16.23)

**Table 1. Socio-demographic characterstics of newly diagnosed type 1DM with DKA and without DKA, 2018 to 2022.**

| Variable | Category | DKA | Non-DKA | TOTAL |
|---|---|---|---|---|
| | | n = 37 (%) | n = 24 (%) | n = 61 (%) |
| Age(year) | 0–4.49 | 8(21.6%) | 3(12.5%) | 11(18%) |
| | 4.5–6.99 | 15(40.6%) | 9(37.5%) | 24(39.4%) |
| | 7–9.49 | 6(16.2%) | 5(20.8%) | 11(18%) |
| | >9.5 | 8(21.6%) | 7(29.2%) | 15(24.6%) |
| Sex | M | 13(35.2%) | 11(45.8%) | 24(39.3%) |
| | F | 24(64.8%) | 13((54.2%) | 37(60.7%) |
| Family history of DM | Yes | 7(18.9%) | 13(54.2%) | 20(32.8%) |
| | No | 30(81.1%) | 11(45.8%) | 41(67.2%) |
| Family income(USD) | <18.02 | 3(8.1% | 1(4.2%) | 4(6.6%) |
| | 18.02–39.64 | 17(46%) | 2(8.3%) | 19(31.1%) |
| | 39.65–61.62 | 7(18.9%) | 2(8.3%) | 9(14.8%) |
| | 61.63–64.87 | 2(5.4%) | 3(12.5%) | 5(8.2%) |
| | >64.87 | 8(21.6%) | 16(66.7%) | 24(39.3%) |
| Parental knowledge of signs and symptoms of DM | Aware | 12(32.4%) | 21(87.5%) | 33(54.1%) |
| | Not aware | 25(67.6%) | 3(12.5%) | 28((45.9%) |
| Infection prior to diagnosis | Yes | 21(56.8%) | 5(20.8%) | 26(42.6%) |
| | No | 16(43.2%) | 19(79.2%) | 35(57.4%) |

showed significant associations and were candidates for multivariate analysis. The odds of developing DKA in females and age groups of above 9.5 years were higher than males and age groups below 9.5 years respectively, however, the association was not significant. Similarly, parental educational level, marital status, occupation and signs, and symptoms of DM that the child had prior to diagnosis did not have a significant association (Table 3).

In multivariate analysis, significant associations were confirmed. Those children whose parents/guardians knew the signs and symptoms of DKA/DM were 93% less likely to develop

**Table 2. Clinical presentations of newly diagnosed type 1DM with DKA, 2018 to 2022.**

| Clinical presentations(n = 37) | |
|---|---|
| Abdominal pain | 10(27%) |
| Dehydration | 12(32.4%) |
| Loss of consciousness | 4(10.8%) |
| Vomiting | 2(5.4%) |
| Fast breathing | 9(24.3%) |
| **Infectious morbidity(n = 21)** | |
| Pneumonia | 7(33.3%) |
| Diarrhea | 6(28.6%) |
| UTI | 4(19%) |
| Tonsilopharyngitis | 3(14.3%) |
| other | 1(4.8%) |
| **Sign and symptoms of DM before diagnosis(n = 61)** | |
| Polyuria | 7(11.5%) |
| Polydipsia | 16(26.2%) |
| Weight loss | 13(21.3%) |
| Polyuria, polydipsia and weigth loss | 16(26.2%) |
| No symptoms | 9(14.8%) |

**Table 3. Bivariable analysis of newly diagnosed type 1 DM, 2018 to 2022.**

| Variables | Category | DM(n = 61) | | OR(95% CI) | P -value |
|---|---|---|---|---|---|
| | | DKA(n = 37) | No DKA(n = 24) | | |
| Age(year) | <2 | 8 | 3 | 0.375(0.017–8.103) | 0.532 |
| | 2–4.49 | 15 | 9 | 0.833(0.041–16.994) | 0.906 |
| | 4.5–6.99 | 6 | 5 | 0.857(0.044–16.851) | 0.919 |
| | 7–9.49 | 8 | 7 | 0.600(0.033–10.822) | 0.723 |
| Sex | M | 13 | 11 | 0.64(0.224–1.827) | 0.405 |
| | F | 24 | 13 | Ref | |
| Family history of DM | Yes | 7 | 13 | 0.197(0.063–0.623) | 0.006* |
| | No | 30 | 11 | Ref | |
| Parental educational level | Can read and write | 14 | 5 | 0.667(0.025–18.059) | 0.810 |
| | Primary education | 12 | 7 | 0.389(0.052–2.924) | 0.359 |
| | Secondary education | 9 | 9 | 0.238(0.030–1.868) | 0.172 |
| | Higher education | 2 | 3 | 0.667(0.087–5.127) | 0.697 |
| Parents knew signs and symptoms of DM/DKA | Yes | 12 | 21 | 0.069(0.017–0.276) | 0.000* |
| | No | 25 | 3 | Ref | |
| Average family income(USD) | <64.87 | 29 | 8 | Ref | |
| | >64.87 | 8 | 16 | 2.261(1.437–3.556) | 0.000* |
| Infection before diagnosis | Yes | 21 | 5 | 4.987(1.532–16.24) | 0.008* |
| | No | 16 | 19 | Ref | |
| Signs and symptoms of DM/DKA that the child had prior to diagnosis | yes | 37 | 15 | 0.158(0.018–1.353) | 0.092 |
| | No | 0 | 9 | Ref | |

DKA at the time of the initial diagnosis of DM (p = 0.017; AOR = 0.070(0.008–0.618)). The presence of a first-degree relative with DM was found to be protective against DKA as an initial presentation of DM (p-value of 0.039, AOR = 0.129, 95% CI (0.019–0.897)) compared to having no first-degree relatives with DM. New-onset DM without DKA is 1.5 times higher in children from families with a high monthly income (p value 0.000, AOR = 1.473, 95% CI (0.679–3.195)) compared to those from families with low income (Table 4).

**Table 4. Bivariable and multivariable analysis of newly diagnosed type 1DM, 2018 to 2022.**

| Variable | Category | DM(n = 61) | | COR(95%CI) | AOR(95%CI) | P-value |
|---|---|---|---|---|---|---|
| | | DKA(n = 37) | No DKA(n = 24) | | | |
| Family history of DM | yes | 7 | 13 | 0.197 (0.063–0.623) | 0.129(0.019–0.897) | 0.039* |
| | No | 30 | 11 | Ref | Ref | |
| Parents knew signs and symptoms of DM/DKA | Yes | 12 | 21 | 0.069(0.017–0.276) | 0.070(.008–0.618) | 0.017* |
| | No | 25 | 3 | Ref | Ref | |
| Average family income(USD) | <64.87 | 29 | 8 | Ref | Ref | |
| | ≥64.87 | 8 | 16 | 2.261(1.43–3.55) | 1.473(.679–3.195) | 0.000* |
| Infection before diagnosis | Yes | 21 | 5 | 4.987(1.53–16.24) | 11.69(1.34–101) | 0.026* |
| | No | 16 | 19 | Ref | Ref | |

Note 1 USD, United States Dollar; DKA, Diabetic Ketoacidosis; DM, Diabetes Mellitus

Note 2

*covariates that showed a statistical significance level (p value <0.2) on bivariate analyses were included in multivariate analyses;

Ref, reference; AOR, Adjusted Odds Ratio; CI, Confidence Interval; COR, Crude Odds Ratio

## Discussion

The frequency of childhood and adolescent Diabetic Ketoacidosis (DKA) as an initial presentation of diabetes mellitus varies worldwide. This retrospective study aimed to review the number of children presenting with DKA at diagnosis and to reveal the background risks for its occurrence in rural and resource-limited settings in Ethiopia. In this study, 60.7% of the children had DKA at diagnosis of diabetes mellitus, a figure notably higher than the 35.8% reported in Addis Ababa 15 years ago [7, 21], but lower than the 78.7% observed in the northern part of Ethiopia, Tigray [8]. The high level of literacy in the community and the availability of a wide range of awareness creation methods (such as screening, posters, flyers, and awareness campaigns) in the capital might explain the sharp decline in the incidence. The model was demonstrated in Germany, where the incidence was reduced from 28% to 16% through the Stuttgart ketoacidosis campaign, which focused on the symptoms of type 1 DM [11]. Moreover, the current study coincided with the peak of the COVID-19 pandemic in Ethiopia, a time when similar studies from Egypt [22], Turkey [23] and a systematic review and meta-analysis across Europe, Asia, and North America [24], observed an increasing trend of newly diagnosed type 1 DM. The rise in the incidence could be due to viral factor-induced new onset Type 1 DM, in which the virus induces pancreatic islet cell and glucose metabolism dysfunction, compounded by reduced health seeking and utilization behavior due to pandemic related fears [24, 25].

In the broader African context, the current prevalence is higher than in Egypt (46%) [16], Sudan (17.6%) [26], and Tanzania (43%) [27], but lower than that in two studies in Nigeria (100%) [10] and (77%) [28]. Outside of Africa, the reported prevalence varies widely: the UK (25%) [29], Germany (20%) [30], Finland (19.4%) [31], Sweden (16%), USA (30%), Canada (18.6%) [3], Kuwait (35.9%), and China (50.1%) [32]. Various factors have been identified for a wide range of variations, including an inverse relationship between DKA and type 1 diabetes mellitus prevalence [33, 34].

The diagnosis of type 1 DM at an average age of 8 years and the frequent occurrence of DKA at a younger age in the current study align with several previous studies conducted in different settings. A combination of genetic predisposition, environmental triggers, hormonal changes, growth and development factors, and delayed recognition of symptoms may contribute to the diagnosis of type 1 diabetes mellitus around this age [12–14, 27, 33]. In contrast to earlier studies in Ethiopia [7, 8, 21], Nigeria [7], Saudi Arabia [13] and Wales [35], the current study observed a higher number of female children presenting with DKA as an initial presentation of type 1 DM than males. This observation is in line with other studies [36]; however, none of the previous studies could explain the increased occurrence in the females. A recent study from England, Germany, Wales, Austria, and the United States was unable to explain female predominance in terms of clinical, biological, and physiological factors [37].

The finding of vomiting accompanied by signs of dehydration, polyuria, polydipsia, and weight loss, and abdominal pain in the present study is consistent with research in northern and central Ethiopia [7, 8, 21, 38] and other studies where they were dominant presentations [3, 16, 27, 32, 39–41]. Additionally, the identification of infectious triggers, particularly pneumonia, diarrheal diseases, and urinary tract infections, as predictors of DKA supports the fact that infections can lead to acute metabolic decompensation and insulin resistance and subsequent development of diabetic ketoacidosis [36]. The 33%comorbidity rate of pneumonia and diarrheal diseases may reflect the underlying immune dysfunction associated with Diabetes. This highlights the critical importance of screening in these cases, as well as the necessity for enhanced preventive measures, such as vaccinations.

A significant finding of the current study was that children whose parents who were unaware of the signs and symptoms of DM were more prone to develop DKA as an initial presentation. Furthermore, the study found that having no family history of diabetes and low monthly family incomes were associated with a diagnosis of DKA at presentation. This finding is in line with studies that have suggested that creating awareness of the signs and symptoms of classic symptoms of DM reduces the incidence of DKA at diagnosis [3, 39, 42–47]. Different strategies have been used to create awareness, such as awareness campaigns on classic symptoms of DM at the local (such as schools, parents/care givers, and health care workers) or nationwide levels. Such methods are effective across countries [11, 45, 48–50] however, some are not [51, 52]. Conversely, the study revealed a notable 32.8% prevalence of family history among newly diagnosed type 1 DM cases, indicating the familial clustering of type 1 DM within this population and providing valuable insight for screening strategies, ultimately reducing DKA at the initial diagnosis.

Results from various studies indicate that high family income is linked to a decreased risk of DKA at diagnosis, although some studies reported no significant effect [36]. When families have a higher income, they are better positioned to seek medical advice without financial constraints-provided they are adequately aware of the signs and symptoms of DM, as shown in the current study.

In agreement with earlier studies [7] and research conducted in Germany [53], the current study shows that despite most parents/guardians having primary and secondary education levels, the education levels of parents do not affect the prevalence of DKA at presentation. However, several studies have shown a protective effect [36, 54–57]. The lack of awareness and information provided on pediatric diabetes or DKA contributed to the insignificant effect of educational level on the occurrence of DKA. Similarly, in line with other studies [54, 58, 59], the current study found no association between parental employment, marital status, and DKA as an initial presentation. The study population, mainly of similar socioeconomic status, and the presence of unmeasured factors such as healthcare utilization, diet, physical activity, and stress levels, may explain the lack of association between parental employment, marital status, and DKA as an initial presentation.

## Study limitations

The study identifies important factors related to DKA development, though its retrospective nature and being a single center might not reflect the condition in the whole population. Short of resources leads to non-identification of biochemical DKA to comply with ISPAD guideline is another limitation.

## Conclusions

In conclusion, the current study indicates that the number of children presenting with DKA at the time of type 1 DM diagnosis was high compared to both national and global studies. The unpredictable course in younger children, masking effect of infectious comorbidities, and low awareness of type 1 DM at the individual level were the main factors. Effective awareness creation methods must be implemented to reduce the burden.

## Supporting information

**S1 File.**
(DOCX)

**S2 File.**
(DOCX)

## Acknowledgments

The acknowledgment goes to all parents and guardians who were willing to provide their consent to participate in the research, and all healthcare workers at the Department of Pediatrics and Child Health, who provided unreserved care for these children.

## Author Contributions

**Conceptualization:** Dinberu Oyamo Oromo.

**Data curation:** Dinberu Oyamo Oromo.

**Validation:** Dinberu Oyamo Oromo.

**Visualization:** Dinberu Oyamo Oromo.

**Writing – original draft:** Dinberu Oyamo Oromo.

**Writing – review & editing:** Dinberu Oyamo Oromo.

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
