## [Decision Letter · Decision Letter 0]

23 Sep 2024

PONE-D-24-27986Pediatric Diabetic Ketoacidosis (PDKA) among newly diagnosed diabetic patients at Dilla University Hospital, Dilla, Ethiopia: prevalence and predictorsPLOS ONE

Dear Dr. Oromo,

Thank you for submitting your manuscript to PLOS ONE. After careful consideration, we feel that it has merit but does not fully meet PLOS ONE’s publication criteria as it currently stands. Therefore, we invite you to submit a revised version of the manuscript that addresses the points raised during the review process. Besides the important reviewrers comments shown below, authors should also address the following points:Authors responded to the question on “Describe where the data may be found” by “Available on reasonable request”. However, according to PLOS ONE publication criteria, authors are required to make all data underlying the findings described fully available, without restriction, and from the time of publication. PLOS allows rare exceptions to address legal and ethical concerns, but these must be cleary explained. Therefore, authors are kindly asked to provide the data supporting study findings (unidentified individual patients’ data) as supplementary file or link to electronic deposit. Authors have to provide in the methodology clear definitions and categorizations for all used variables, including age, family income, parental employment, parental education status, parental marital status, and awareness. In particular, they should clarify how awareness was assessed and categorized?Authors should provide more details on data collection methods, including whether the records were paper or electronic, who collected the data, any double check?Authors should present all results of bivariate analysis for all independent factors, not only those with significant results.Authors should be careful in interpreting OR. OR cannot be interpreted as probability (percentage) or like relative risk.In discussion, please, ensure adequate discussion for most important findings, including also association with higher income.Please, acknowledge study limitations, including retrospective study design, realtively small sample size, and being single center study.In abstract: authors report inadequate knowledge of DM and absence of family history as risk factors for DKA among children with newly diagnosed T1DM, but they report the AOR of 0.07 and 0.129, which are derived from their statistical analysis for the association of ADEQUATE knowledge and PRESENCE of family history of DM. This would be confusing for the readers when speaking about risk factors but providing OR less than one which are interpreted as protective. Authors should revise this part.In abstract, authors should also report the high income as a significant predictor. The term sex would be preferred if you refer to biological variable, but gender would be mor appropriate if you refer to social or cultural variable.

We look forward to receiving your revised manuscript.

Kind regards,

Elsayed Abdelkreem, MD, PhD

Academic Editor

PLOS ONE

2. In the online submission form, you indicated that [Insert text from online submission form here].

Reviewers' comments:

Reviewer's Responses to Questions

**Comments to the Author**

1. Is the manuscript technically sound, and do the data support the conclusions?

Reviewer #1: Partly

Reviewer #2: Partly

Reviewer #3: Yes

2. Has the statistical analysis been performed appropriately and rigorously? 

Reviewer #1: I Don't Know

Reviewer #2: Yes

Reviewer #3: Yes

3. Have the authors made all data underlying the findings in their manuscript fully available?

Reviewer #1: Yes

Reviewer #2: Yes

Reviewer #3: Yes

4. Is the manuscript presented in an intelligible fashion and written in standard English?

Reviewer #1: Yes

Reviewer #2: Yes

Reviewer #3: Yes

5. Review Comments to the Author

Reviewer #1: This is an interesting article in which the author shows the prevalence and predictors of diabetic ketoacidosis in children in a centre in Ethiopia. Since few articles have this information, it gives us interesting data. However, the work lacks essential data, primarily related to the definition of diabetic ketoacidosis.

In the introduction, a new paper about the prevalence of DKA in children in Ethiopia from 2019 should be included in the references (Assefa B. et al. Incidence and predictors of diabetic ketoacidosis among children with diabetes in west and east Gojjam zone referral hospitals, northern 2019. Ital J Pediatr. 3;46(1):164. PMID: 33143741.)

According to the 2022 ISPAD guidelines (Glaser N. et al. ISPAD clinical practice consensus guidelines 2022: Diabetic ketoacidosis and hyperglycemic hyperosmolar state. Pediatr Diabetes. 2022 Nov ;23(7):835-856. doi: 10.1111/pedi.13406. PMID: 36250645.) the diagnosis of DKA is based on the triad of hyperglycemia, ketosis and metabolic acidosis; however, specific biochemical criteria used to define DKA vary in different parts of the world and among different research studies. All three biochemical criteria are required to diagnose DKA:

• Hyperglycemia (blood glucose >11 mmol/L [200 mg/dl])

• Venous pH < 7.3 or serum bicarbonate < 18 mmol/L

• Ketonemia or ketonuria

In this paper, the group of children with DKA was defined based on the clinical picture, blood glucose and ketones, and the clinical picture, while venous pH and serum bicarbonates were not determined. The above is very unreliable for definition. The question is how many more children with newly diagnosed T1DM should be in the DKA group if pH and bicarbonates were determined (line 85).

The Dilla University Ethical Commission's approval number needs to be listed. I suggest entering it (line 108).

Figure 1 is referenced in the results, but it is not present in the work. This figure must be included, as it could provide valuable visual context to the findings, enhancing the reader's understanding of the research.

The references in this article must be revised and updated. For instance, references 3 and 16 are identical and should be corrected. This attention to detail will ensure the accuracy and thoroughness of the research, instilling confidence in the reader.

Reviewer #2: The manuscript titled “Pediatric Diabetic Ketoacidosis (PDKA) among newly diagnosed diabetic patients at Dilla University Hospital, Dilla, Ethiopia: prevalence and predictors” aimed to know the prevalence of diabetic ketoacidosis among patients with newly diagnosed diabetes mellitus and identify avoidable risk factors.

The study raised the importance social issues, including the literacy and the family income

However, it was carried on during the period 2018-2022, and the COVID era started 2019. All patients were influenced by COVID which reflected their presentations and even the delay to seek medical advice

1. Methodology:

- Criteria of DKA diagnosis: why not mentioned the Venous blood gases results (HCO3 & Ph)

- Assessment of knowledge: items to be assessed not mentioned clearly

- Assessment of income: to be translated in Dollars and according to the world bank classification”

2. the tables:

- n to write the total numbers

- comparing the patients yearly will clarify the effect of COVID during this era “to be added”

3. the discussion:

- to mention the COVID and its effect on the patient’s presentation

Reviewer #3: The manuscript aims to identify the prevalence of DKA among newly type 1 diabetes cases at one single-center in Ethiopia and detect potential predictors

The strengths of this manuscript rely on the interest to address a question that has not been sufficiently studied in the region. The manuscript is well written and the conclusions are clear based on findings.

The main limitation is that biochemical parameters considered to diagnose DKA according to ISPAD guidelines have not been assessed (only urine ketones) and thus the accuracy of the findings may be uncertain. In addition, data refer to a one-single center and may not be representative of country or region reality.

Some points to consider:

Abstract:

-the term “polysymptoms” is inespecific and likely the abstract would benefit of reconsidering this sentence.

-as the manuscript is based on retrospective data the high proportion of DKA in children with newly diagnosis of type 1 diabetes cannot directly be attributed to the described factors. Please reconsider this sentence

Manuscript:

Line 44: the first cite refers to ISPAD guidelines 2007. Last version of these guidelines was 2022 and thus it is outdated. Please consider to update the cite.

Line 60: the term “Materials and methods” should be replaced by “Participants and Methods”

Line 77: the term “gender” is preferable to “sex”. Similarly in other parts of the manuscript.

Other comments that should be addressed in the discussion

>20% of family history in newly diagnosis is a very high rate. Please discuss it.

33% of pneumonia co-morbidity is a very high rate. Please discuss it.

Please specify what polysymptoms is.

There are not data regarding mortality rates in newly diagnosis type 1 diabetes. Please discuss it.

Biochemical parameters considered to diagnose DKA according to ISPAD guidelines have not been assessed (only urine ketones). Please discuss it.

6. PLOS authors have the option to publish the peer review history of their article (what does this mean?). If published, this will include your full peer review and any attached files.

Reviewer #1: No

Reviewer #2: **Yes: **Mona Karem Amin

Reviewer #3: No

---

## [Author Response · Author response to Decision Letter 0]

28 Oct 2024

Dinberu Oyamo, MD, Pediatrician

Dilla University

dinbe80@gmail.com

Elsayed Abdekreem, MD, PhD

Academic Editor

PLOS ONE

Dear Dr. Elsayed Abdelkreem, 

Thank you for your letter and the reviewers’ comments regarding the manuscript titled “Pediatric Diabetic Ketoacidosis (PDKA) among newly diagnosed diabetic patients at Dilla University Hospital: prevalence and predictors” (Manuscript Number: PONE-D-24-27986). I would like to extend my gratitude to all the reviewers who took the time to review the manuscript.

All the comments provided are valuable and will significantly aid in revising and improving the paper. Corrections have been made to the manuscript based on the comments from the academic editor and reviewers. As per your instructions, a marked-up copy with track changes as well as an unmarked version of the revised manuscript, have been uploaded. In addition, the manuscript has been thoroughly revised to comply with PLOS ONE guidelines.

Below is a point-by-point response to each comment raised by the academic editor and reviewers. Reviewers’ comments are written in italic.

Thank you for inviting to submit a revised manuscript, and I hope the revisions made meet with your approval.

Sincerely,

Dinberu Oyamo, MD

Corresponding Author

Response to Academic Editor Comments:

Dear academic editor,

Thank you very much for handling the manuscript and thoroughly reviewing it. I hope that you will find the responses and revised manuscript satisfactory.

Sincerely,

1)Comment: 

Authors responded to the question on “Describe where the data may be found” by “Available on reasonable request”. However, according to PLOS ONE publication criteria, authors are required to make all data underlying the findings described fully available, without restriction, and from the time of publication. PLOS allows rare exceptions to address legal and ethical concerns, but these must be clearly explained. Therefore, authors are kindly asked to provide the data supporting study findings (unidentified individual patients’ data) as supplementary file or link to electronic deposit.

Response: 

Thank you for your valuable feedback regarding data availability. I apologize for the oversight in the initial submission. The data supporting the study findings have been provided within the manuscript and as a supplementary file.

2) Comment:

 Authors have to provide in the methodology clear definitions and categorizations for all used variables, including age, family income, parental employment, parental education status, parental marital status, and awareness. In particular, they should clarify how awareness was assessed and categorized?

Response: 

Thank you for your valuable suggestion regarding the definitions and categorizations of the variables used in the study. In response to your feedback, clear definitions and categorizations for all specified variables have been incorporated into the participant and method sections of the manuscript (please refer page no 5, line no 90).

3) Comment: 

 Authors should provide more details on data collection methods, including whether the records were paper or electronic, who collected the data, any double check?

Response:

Thank you for the suggestion regarding data collection methods. Additional details on the data collection procedures, including the format of records (paper or electronic), the individuals responsible for data collection, and the measures taken for double-checking the data, have been included in the manuscript(Please refer page no 5,line no 77 to 89).

4) Comment: 

Authors should present all results of bivariate analysis for all independent factors, not only those with significant results

Response: 

Thank you for your valuable suggestion. The Results of bivariate analysis of all independent variables are provided under the table titled” Bivariable analysis of newly diagnosed type 1 DM, 2018 to 2022” (please see table 3, under result section).

5) Comment: 

Authors should be careful in interpreting OR. OR cannot be interpreted as probability (percentage) or like relative risk.

Response:

Thank you for highlighting this important point. Appropriate adjustments have been made in the manuscript to clarify that OR reflect the odds of an event occurring in one group relative to another, rather than a direct probability or percentage. Specific notes have been added in the results and discussion sections on (please refer page no 12, line no 197 to 204).

6) Comment: 

 In discussion, please, ensure adequate discussion for most important findings, including also association with higher income.

Response: 

Thank you for your valuable suggestion regarding the discussion of key findings. Additional content has been incorporated to enhance the discussion, including income (Please refer page no 14 ,15 and 16, line numbers 223 to 229 and 267 to 275).

7) Comment: 

Please, acknowledge study limitations, including retrospective study design, realtively small sample size, and being single center study.

Response: 

Thank you for your suggestion to acknowledge the study limitations. Acknowledgement of the study’s limitations has been added to the manuscript. (Please refer page no 16,line no 287)

8) Comment: 

 In abstract: authors report inadequate knowledge of DM and absence of family history as risk factors for DKA among children with newly diagnosed T1DM, but they report the AOR of 0.07 and 0.129, which are derived from their statistical analysis for the association of ADEQUATE knowledge and PRESENCE of family history of DM. This would be confusing for the readers when speaking about risk factors but providing OR less than one which are interpreted as protective. Authors should revise this part. In abstract, authors should also report the high income as a significant predictor.

Response: 

Thank you for your insightful feedback regarding the interpretation of odds ratios (AOR). The statements have been revised to clarify the protective nature of adequate knowledge and family history of diabetes, as well as to include high income as a significant predictor. The revised section now read: “The presence of adequate knowledge of signs and symptoms of DM (AOR=0.07, 95%CI 0.019-0.0897, P value 0.017) and a family history of DM (AOR=0.129 95%CI 0.019-0.897, P value 0.039) were protective factors against DKA at the initial diagnosis of DM. Moreover, new-onset type 1 DM without DKA was 1.5 times higher in children from families with a high monthly income (AOR=1.473, 95% CI 0.679-3.195 p value 0.000) compared to those from families with low income. The presence of an infection prior to DKA (AOR=11.69,95%CI 1.34-10.1,P value 0.026) was associated with the diagnosis of DKA at the initial presentation of DM.” (Please refer page no 2, line no 30, under the result section of the abstract).

9) Comment: 

The term sex would be preferred if you refer to biological variable, but gender would be mor appropriate if you refer to social or cultural variable.

Response: 

Thank you for your comments to clarify the term “sex”.The term “sex” is consistently used throughout the manuscript to refer specifically to the biological variable, distinguishing it from “gender”, which refers to cultural/social variable.

—The end—

Response to Reviewers 1, 2 and 3

Response to Reviewer #1:

Dear reviewer #1,

Thank you very much for reviewing the manuscript and for your crucial comments and suggestions. I hope that you will find the responses and revised manuscript satisfactory.

Sincerely,

1) Comment:

This is an interesting article in which the author shows the prevalence and predictors of diabetic ketoacidosis in children in a centre in Ethiopia. Since few articles have this information, it gives us interesting data. However, the work lacks essential data, primarily related to the definition of diabetic ketoacidosis.In the introduction, a new paper about the prevalence of DKA in children in Ethiopia from 2019 should be included in the references (Assefa B. et al. Incidence and predictors of diabetic ketoacidosis among children with diabetes in west and east Gojjam zone referral hospitals, northern 2019. Ital J Pediatr. 3;46(1):164. PMID: 33143741.)

Response: 

Thank you for your positive feedback on the relevance of the research. The concern about the lack of essential data related to the definition of DKA is acknowledged and addressed by providing a comprehensive explanation of the criteria used to define DKA based on the 2022 ISPAD guideline in the revised manuscript(please refer page no 5, line no 90). Additionally, the suggested reference by Assefa B et al has been included in the introduction section (please refer page no 3, line no 54,and reference no 16).Furthermore ,the suggested reference, Assefa B et al ,primarily focus on known type 1 DM children who were on follow up excluding newly diagnosed type 1 DM children with DKA diagnosed during the study period.

2) Comment: 

The Dilla University Ethical Commission's approval number needs to be listed. I suggest entering it (line 108).

 Figure 1 is referenced in the results, but it is not present in the work. This figure must be included, as it could provide valuable visual context to the findings, enhancing the reader's understanding of the research. 

The references in this article must be revised and updated. For instance, references 3 and 16 are identical and should be corrected. This attention to detail will ensure the accuracy and thoroughness of the research, instilling confidence in the reader.

Response: 

Thank you for pointing out these issues and for your valuable suggestions. As per your suggestion, the ethical number has been added (please refer page no 7, line no 145), and Figure 1is provided in the manuscript and uploaded as a TIF file. Additionally, the duplicate references (references 3 and 16) have been corrected .Reference 16 has been removed and replaced with an additional relevant reference (please refer page no 19, line no 348, reference no 16). 

—The end—

Response to Reviewer #2:

Dear reviewer #2,

Thank you very much for reviewing the manuscript and for your crucial comments and suggestions. I hope that you will find the responses and revised manuscript satisfactory.

Sincerely, 

1)  Comment: 

The study raised the importance social issues, including the literacy and the family income.However, it was carried on during the period 2018-2022, and the COVID era started 2019. All patients were influenced by COVID which reflected their presentations and even the delay to seek medical advice

Response:

 Thank you for your positive feedback on the relevance of this research. The impact of COVID-19 on patients during the study period is indeed significant and has been acknowledged and discussed in the revised manuscript(please refer page no 14, line no 223 to 229 and reference numbers 58, 59, 60 and 61).

2) Comment: 

Methodology:- Criteria of DKA diagnosis: why not mentioned the Venous blood gases results (HCO3 & Ph)- Assessment of knowledge: items to be assessed not mentioned clearly- Assessment of income: to be translated in Dollars and according to the world bank classification”

Response: 

Thank you for your valuable comments and suggestions regarding the methodology. The comments have been accepted. A comprehensive explanation of the criteria used to define DKA based on the 2022 ISPAD guideline has been added in the revised manuscript in the method section(please refer page no 5,line no 90).The assessments of Knowledge and the items used have been included in the method section. Income assessment (translated in to dollars) has been included in the method section. (please refer page no 6,line no 104 an d 113)

3) Comment:

 the tables:- n to write the total numbers- comparing the patients yearly will clarify the effect of COVID during this era “to be added”

Response: 

Thank you for your valuable suggestions. The recommendations have been accepted, and the total numbers (n) have been added in the tables (please refer tables 1 to 4).

4) Comment:

 the discussion:- to mention the COVID and its effect on the patient’s presentation

Response: 

Thank you for your valuable suggestion regarding the discussion of COVID-19’s impact on patient presentations. This suggestion has been accepted, and a section discussing the effects of COVID-19 on the patient population has been added to the discussion. Relevant references have also been included to support this addition (please refer page no 14, line no 223 to 229 and reference numbers 58, 59, 60 and 61).

—The end—

Response to Reviewer #3:

Dear reviewer #3,

Thank you very much for reviewing the manuscript and for your crucial comments and suggestions. I hope that you will find the responses and revised manuscript satisfactory.

Sincerely, 

1) Comment:

 The manuscript aims to identify the prevalence of DKA among newly type 1 diabetes cases at one single-center in Ethiopia and detect potential predictorsThe strengths of this manuscript rely on the interest to address a question that has not been sufficiently studied in the region. The manuscript is well written and the conclusions are clear based on findings.The main limitation is that biochemical parameters considered to diagnose DKA according to ISPAD guidelines have not been assessed (only urine ketones) and thus the accuracy of the findings may be uncertain. In addition, data refer to a one-single center and may not be representative of country or region reality

Response:

Thank you for your valuable feedback. The recognition of the study’s contribution to addressing a gap in the region is greatly appreciated. Regarding the diagnosis of DKA, the limitation concerning the assessment of biochemical parameters has been acknowledged. A definition of DKA in resource-limited settings, adopted from ISPAD guidelines, which are used across all over the country, has been included in the method section of the revised manuscript (please refer page no5, line no 90).Furthermore, the study limitations, including being a single center, have been stated in the revised manuscript(please refer page no 16 ,line no 287)

2) Comment: 

Abstract:-the term “polysymptoms” is inespecific and likely, the abstract would benefit of reconsidering this sentence.-as the manuscript is based on retrospective data the high proportion of DKA in children with newly diagnosis of type 1 diabetes cannot directly be attributed to the described factors. Please reconsider this sentence

Response: 

Thank you for your valuable feedback regarding the abstract. The term”polysymptoms” has been revised to “polyuria, polydipsia and associated weight loss” for greater specificity(please refer page no 2 line no 29,and page no10 line no 177).Additionally, the phrase “… is attributed to” has been changed to”…is associated with” to reflect the retrospective nature of the data more accurately(please refer page no 3, line no 40).

3) Comment:

Manuscript: Line 44: the first cite refers to ISPAD guidelines 2007. Last version of these guidelines was 2022 and thus it is outdated. Please consider to update the cite.Line 60: the term “Materials and methods” should be replaced by “Participants and Methods”Line 77: the term “gender” is preferable to “sex”. Similarly in other parts of the manuscript.

Response: 

Thank you for your valuable observations and suggestions. The reference to the ISPAD guidelines has been updated to the most recent version (2022) (please refer page no 17, line no 307-reference no1).The term “Material and Method “has been replaced with “Participants and Methods” as suggested (Please refer page no 4, line no 63). The term “Sex” has been used throughout the manuscript to ensure consistency and to refer specifically to the biological variable.

4) Comment:

 Other comments that should be addressed in the discussion

a) >20% of family history in newly diagnosis is a very high rate. Please discuss it.

 b)33% of pneumonia co-morbidity is a very high rate. Ple

---

## [Decision Letter · Decision Letter 1]

11 Nov 2024

Pediatric Diabetic Ketoacidosis (PDKA) among newly diagnosed diabetic patients at Dilla University Hospital, Dilla, Ethiopia: prevalence and predictors

PONE-D-24-27986R1

Dear Dr. Oromo,

We’re pleased to inform you that your manuscript has been judged scientifically suitable for publication and will be formally accepted for publication once it meets all outstanding technical requirements.

Kind regards,

Elsayed Abdelkreem, MD, PhD

Academic Editor

PLOS ONE

Additional Editor Comments (optional):

Reviewers' comments:

Reviewer's Responses to Questions

**Comments to the Author**

1. If the authors have adequately addressed your comments raised in a previous round of review and you feel that this manuscript is now acceptable for publication, you may indicate that here to bypass the “Comments to the Author” section, enter your conflict of interest statement in the “Confidential to Editor” section, and submit your "Accept" recommendation.

Reviewer #1: All comments have been addressed

Reviewer #3: All comments have been addressed

2. Is the manuscript technically sound, and do the data support the conclusions?

Reviewer #1: Yes

Reviewer #3: Partly

3. Has the statistical analysis been performed appropriately and rigorously? 

Reviewer #1: I Don't Know

Reviewer #3: Yes

4. Have the authors made all data underlying the findings in their manuscript fully available?

Reviewer #1: Yes

Reviewer #3: Yes

5. Is the manuscript presented in an intelligible fashion and written in standard English?

Reviewer #1: Yes

Reviewer #3: Yes

6. Review Comments to the Author

Reviewer #1: (No Response)

Reviewer #3: Thanks for your comments and the modifications in the manuscript. The authors have gone through my criticism. No further comments from my side at this moment.

7. PLOS authors have the option to publish the peer review history of their article (what does this mean?). If published, this will include your full peer review and any attached files.

Reviewer #1: No

Reviewer #3: No

---

## [Editor Report · Acceptance letter]

15 Nov 2024

PONE-D-24-27986R1 

PLOS ONE

Dear Dr. Oromo, 

I'm pleased to inform you that your manuscript has been deemed suitable for publication in PLOS ONE. Congratulations! Your manuscript is now being handed over to our production team.

Kind regards, 

on behalf of

Dr. Elsayed Abdelkreem 

Academic Editor

PLOS ONE